# Multiphase Flow Production Enhancement Using Drag Reducing Polymers

**DOI:** 10.3390/polym15051108

**Published:** 2023-02-23

**Authors:** Abdelsalam Alsarkhi, Mustafa Salah

**Affiliations:** Department of Mechanical Engineering, Center for Integrative Petroleum Research, King Fahd University of Petroleum and Minerals, Dhahran 31261, Saudi Arabia

**Keywords:** drag reducing polymers, multiphase flow, production enhancement, flow pattern transition, disturbance waves

## Abstract

This paper presents a comprehensive experimental investigation concerning the effect of drag reducing polymers (DRP) on enhancing the throughput and reducing the pressure drop for a horizontal pipe carrying two-phase flow of air and water mixture. Moreover, the ability of these polymer entanglements to damp turbulence waves and changing the flow regime has been tested at various conditions, and a clear observation showed that the maximum drag reduction always occurs when the highly fluctuated waves were reduced effectively by DRP (and that, accordingly, phase transition (flow regime changed) appeared. This may also help in improving the separation process and enhancing the separator performance. The present experimental set-up has been constructed using a test section of 1.016-cm ID; an acrylic tube section was used to enable visual observations of the flow patterns. A new injection technique has been utilized and, with the use of different injection rates of DRP, the results have shown that the reduction in pressure drop occurred in all flow configurations. Furthermore, different empirical correlations have been developed which improve the ability to predict the pressure drop after the addition of DRP. The correlations showed low discrepancy for a wide range of water and air flow rates.

## 1. Introduction and Literature Review

Drag reducer chemicals are high molecular weight polymers (greater than $2\times 10^{6}$). Typical species include polyacrylamides and both natural and Xanthan gums. Their mode of action is believed to be by reducing turbulent eddies and extending the laminar boundary layer at the pipe wall and they are considered to be effective under turbulent conditions.

During the transportation of the multiphase (gas-liquid) in the pipelines industry, several flow regimes might form, leading to a large pressure gradient. To reduce the frictional pressure drop, different techniques have been proposed in the literature. Addition of a few parts per million (ppm) of polymers liquid in the pipe is one way to achieve good drag reduction and reduce the frictional pressure losses.

The concept of adding high molecular weight long-chain polymers into a single-phase liquid flow was first published by Tom [1] and is known as Tom’s phenomenon. In Tom’s work, high reduction was observed on the frictional resistance at the pipe wall which finally leads to the possibility of increasing the pipeline capacities and flow rates.

For the gas-liquid flow in pipes, the effect of the drag reducing polymers on the existing system have been investigated experimentally and several scientific researches have been published. Oliver et al. [2] were the first to investigate the effect of drag reducing polymers in gas-liquid flows using 1.3% polyethylene (PEO) aqueous solution and air. They reported that the liquid in the slug flow where the wave was absorbed gave smooth liquid film.

Al-Sarkhi et al. [3] studied the drag reducing polymers on air-water flow in horizontal pipes, they found that the DRA destroys the turbulent waves which affect the flow rates and the pressure of the system. The maximum drag reduction obtained was about 48% for annular flow configuration. The discussion has been carried out about the effectiveness of the drag reduction agent which is depend on the way of DRA been introduced into the regime, they suggested an injection of a well mixed master solution to the film in order to have good distribution along the pipe circumference.

Soleimani et al. [4] examined the influence of adding polymers on the pseudo slug flow and the transition to slug flow patterns for the air-water two phase configuration using pipe diameter of 2.54 cm. They studied the effect of the polymer concentration on the pressure gradient for different superficial liquid velocities ($U_{\mathrm{sl}}$) and they also noted that the decrease in the pressure gradient is not monotonic with the polymer ppm since the polymers will enlarge the liquid holdup while decreasing the interfacial friction which have an opposite effect on the pressure drop. Therefore an increase or decrease could be realized.

Stratified flow configuration and its transition to slug flow in small pipes may exhibits some complicity than the one flows in large diameters since the interface between gas and the liquid is hidden by the large waves which touch the pipe top wall. Baik et al. [5] investigated the effect of the drag reducing polymers on these waves at high and low superficial gas velocity, they reported that the wave amplitude decreased dramatically using the polymer solution and a drag reduction of about 42% was noted.

Fernandes et al. [6] conducted an experimental study using high molecular weight poly-alpha-olefin polymers on two phase flow(gas-condensate flow) that operate in annular flow regime, they developed a mechanistic model and comparative study by applying the DRA on similar experimental loop of Al-Sarkhi et al. [3] to show the applicability of their model and its limitation. The error between the model and the experimental data was 5%, finally they concluded that as the pipe diameter increases, the drag reduction increases due to the reduction of the entrainment.

Many researchers in the literature continued delivering numerous empirical correlations that help in evaluating the pressure drop occurs within multiphase flows, with respect to various operational conditions those models have been developed. Recently, Al-sarkhi et al. [7] developed two correlations for the friction factor (based on the asymptotic value of drag reduction) for a wide range of pipe diameter from 0.019 to 0.0953 m and using the results of the published data of air–liquid annular flows and liquid–liquid flows to realize the capability of the prediction for any flow pattern with the presence of DRP in pipes.

Al-sarkhi [8] investigated the influence of mixing technique of a drag reducing polymer and the way it is introduced to the gas-liquid annular flow on the percentage of drag reduction by DRP. Effect of different master solution (the injected liquid polymers) concentrations were studied. Al-sarkhi [9] published a very extensive literature review of drag reduction by polymers in gas-liquid and liquid-liquid flows in pipes. In this work, the mechanisms of drag reduction proposals were discussed and the need for further research in this area were identified.

Wang et al. [10] used direct numerical simulation method to investigate the gas liquid drag reducing cavity flow using the volume of fluid and level set method. It was concluded that a high concentration of polymers enhances the drag reduction.

The novelty of the present work is that the experiments were conducted in a 1.016-cm ID stainless steel tube in which the experimental data is rarely exited in open literatures. Moreover, different flow regimes were tested and new empirical correlations are developed which enables predicting the pressure drop and friction factor after the addition of DRP compared to gas-liquid without DRP to give an estimate of the amount of drag reduction.

## 2. Description of Experimental Setup and Procedure

The experimental flow loop depicted in Figure 1 below is designed to investigate the influence of the DRP additives on the flow behavior of liquid and air mixture. The loop comprises two 200 liter barrels for water and an instrument air connection for the air supply. The flow rate of the feed streams is measured and can be adjusted using regulating valves. The additive is added to the flow system via a nozzle into the mixed fluid stream via diaphragm pump.

The feed pumps for the liquids (water) are rotary pumps equipped with axial face sealings. Water, and air can be separated in the separator or using cyclone and separator which are connected to the outlet of the test section. However, in the present paper only air and water are used as the two phases but the loop has the capability of having three phases air, water, and oil.

The test section is made of stainless-steel tube with an outer diameter of 1.27 cm and an inner diameter of 1.016 cm. its total length is approximately 5 m divided into two straight horizontal sections separated by elbows (90-degree elbow). The horizontal sections are equipped with differential pressure transducer to measure the pressure drop inside the test section along a distance of 1.5 m. At the end of the test section an acrylic section of 20 cm long allows the visible inspection of the flow behavior. After having passed the test section, the fluid can be directed to the phase separator where water and air can be separated by gravity or alternatively to the cyclone whose outlet which is connected to the phase separator.

### 2.1. Preparation of the Polymer Solution

A polymer in a powder format is mixed with water in rotating magnetic mixer at low speed in order to avoid polymer shear degradation. Then rotation is stopped when the mixture completely dissolved in the water and having a conglomerated consistency. The mixing process may take several hours and sometimes a heat addition up to 50 °C was used to accelerate the solubility. Water and polymer specifications are given in Table 1 and Table 2. Table 2 shows data from the manufacturer product sheet indicating the viscosity of the DRP at the specific concentration. It is worth to be mentioned here that at 100 PPM DRP in water which is the maximum concentration used in the present experiments neither the viscosity of the water with 100 ppm DRP nor the density or surfactant will be affected by the DRP presence in such a small amount.

### 2.2. System Operation

The generated air-water two phase flow is circulated through the flow loop using a vertical centrifugal pump that can provide a maximum flow rate of 40 L/min of water. On the other hand, the air is introduced to the system (from the laboratory main source) using a pressure regulator (with a maximum 16 bar inlet pressure and maximum 10 bar outlet pressure) connected at the inlet of the compressed air. A thermal mass flow rate measures the air flow in range from (0–150) L/min. The flow rate of the water is measured using an electromagnetic flow meter for flow range up to 40 L/min, a check valve is connected after the flow meter to prevent back flow of water.

## 3. Results and Discussions

### 3.1. Effect of DRP on Frictional Pressure Drop

In this study, the effect of adding the DRP on frictional pressure gradient for air-water mixture has been tested for a wide range of liquid and gas flow rates, the liquid flow rate starts from 3 to 25 L/min while gas flow is up to 70 L/min. The corresponding pressure drop has been recorded for the entire range with and without the DRP.

Figure 2 provides a clear comparison for the frictional pressure drop reported with different liquid superficial velocities (the points symbols stand for pressure drop after the injection of 0.6 L/min DRP and the lines represent pressure drop without adding the DRP). It is indicated that from this figure as the gas superficial velocity increases from 2.6 to 4.11 m/s the pressure drop increases accordingly, one possible justification for this behavior is that once the gas velocity increases an additional pressure loss in the mixture of the two-phase flow appear due to the disturbance in the liquid flow caused by the gas.

### 3.2. Effect of DRP in Two-Phase Flow Pattern Transition

The observed flow patterns gas-liquid two phase with and without the addition of 40 ppm Drag Reducing polymer (DRP) results are illustrated in Appendix A. As can be seen, most of the flow regimes were changed with DRP except the smooth stratified. The minimum percentage of drag reduction was in the stratified flow regime and the maximum was for the slug flow when it is changed to wavy stratified after the addition the DRP.

#### 3.2.1. Stratified and Stratified Wavy Flow Regimes

The reductions in the role waves and ripples have been realized and the flow has become more stable. Furthermore, the range of the smooth stratified flow pattern increased primarily at the transition region between slug and stratified wavy flows.

As it can be seen from Figure 3 that the frictional pressure gradient increases significantly as the dimensionless superficial velocity increases, which is mainly due to the increase in gas superficial velocity that adds more disturbance to the gas liquid interface.

A minimal effect of DRP has been noted for the stratified regime due to uniform and quite stable interface between gas and liquid (air-water). Also, there is no clear transition effects from wavy stratified to stratified flow pattern. However, the role of the waves and their intensity have been damped further with the presence of DRP. Also, as emphasized by Baik et al. [5], the DRP effectively reduced the wave amplitude and delayed transition to slug flow regime. Figure 4 and Figure 5 depict the stratified and wavy stratified flows before and after adding the DRP.

#### 3.2.2. Annular and Wavy Annular Flow Patterns

Annular and wavy annular flow patterns have been studied to show the effectiveness of adding a small concentration of drag reducing polymer. Figure 6 illustrates how the DRP can reduce the pressure drop at various gas superficial velocities. It can be seen clearly that the DRP was able to suppress the waves at the bottom of annular film for all gas flow rates been studied. Thus, a drag reduction has been observed.

Moreover, the transition from wavy annular regime to stratified wavy occurred with the addition of only 40 ppm of DRP. Appendix A summarizes the ranges at which these transitions have been observed, and the maximum drag reduction obtained for the annular and wavy annular region was 48%, this effectiveness decreases as more waves propagate at the annular liquid film.

Taylor et al. [11] divided the annular flow regime into three distinct regions according to liquid film disturbance, first region in which the wave starts to form then augments more in region two, and finally the wave oscillations go down in third region. The overall frequency of the interfacial waves decreases as far as it move downstream Zhao et al. [12]. The energy associated with forming these waves always results in reduction in the total pressure, thus using DRP to damp and delay these oscillations lead to a reduction in pressure drop. Figure 7 and Figure 8 represent typical features of annular and wavy annular flow regimes.

The effectiveness of the drag reducing polymers is very sensitive to the way that the DRP been introduced to the system (Al-Sarkhi et al. [3]), in the present study a diaphragm pump has been utilized to inject the polymer into liquid film of the annular flow to avoid polymer molecules breakup. Figure 7 and Figure 8 indicated that the wavy annular flow shifted slightly to stratified wavy regime as DRP injected.

It should be noted that drag reducing polymers acting to stabilize the liquid film by damping disturbance waves at the gas liquid interface, thus a reduction in pressure drop occurs and also an increase in the mean liquid thickness could be observed, this realization in a good agreement with Spedding et al. [13] and Thwaites et al. [14] findings for the annular flow regime.

#### 3.2.3. Dispersed Bubbly Flow Regime

As the liquid superficial velocity further increases the dispersed bubbly regime would be a possible candidate and generally this flow pattern characterized by small bubbles introduced as a discrete particle in the liquid continuous phase. Figure 9a shows the typical behavior of the bubbly flow. The performance of the DRP has been examined for this type of flow; Figure 10 exhibits the variation of the pressure drop with the dimensionless superficial velocity with and without the DRP. The results are reported in Appendix B, it can be seen that the maximum drag reduction percentage occurred was about 55% and the flow changed slightly to pseudo slug flow regime; these changes were limited up to 5 m/s of gas superficial velocity.

The onset of transition to pseudo slug flow is clearly indicated in Figure 9b with the presence of 40 ppm DRP. The mechanism of the transition is that; with these polymers the separated bubbles tends to coalesce together forming gas pseudo slugs, due to the decrease in the level of turbulence which contribute in keeping the air bubble dispersed in the liquid.

#### 3.2.4. Slug and Pseudo Slug Flow Regimes

A distinctive study has been carried out to examine the effect of the DRP on the characteristics of slug and pseudo slug flows utilizing two polymer concentrations namely 40 and 100 ppm; to show the effectiveness of the DRP in changing the flow patterns at low and relatively high concentrations.

Appendix B and Appendix C articulate the frictional pressure gradient with and without adding DRP of 40 and 100 ppm, respectively. It is noted that adding 40 ppm DRP could results in a decrease of turbulence wave’s intensity and slug frequency with no clear transition from the slug to the stratified wavy regime. The inception of this transition is illustrated with the presence of adding 100 ppm (Figure 11).

As seen from Figure 11 that the pressure drop reduced more in the case of 100 ppm and the maximum Drag Reduction effectiveness reported in the case of 40 ppm was 53%, and 66% for a situation where 100 ppm added to the flow.

It should be noted that the transition from Slug to Stratified wavy flow has been observed for all of the range studied with addition of 100 ppm concentration unlike the case where no transition noted with utilizing only 40 ppm.

The effectiveness of DRP on the pseudo slug regime is exhibited in Appendix D and Appendix E. Here, the possible transition to wavy annular flow started earlier when 40 ppm has been added. Also it has been realized that more disturbance appears as the gas flow rate increases, and there is no changes in the characteristics of pseudo slug regime is observed, though the DRP only acts to decrease the turbulence intensity at the gas liquid interface which is totally support the claims of More et al. [15].

The maximum effectiveness reported was about 41% and 64% for the case of 40 and 100 ppm, respectively. Figure 12 shows the variation of pressure drop with the dimensionless superficial velocity, it is clearly indicated that the pressure drop has been reduced further more in case of adding 100 ppm DRP concentration for both slug and pseudo slug regimes. Increasing the DRP concentration even more up to 100 ppm enhanced the transition to Wavy annular for the whole superficial gas velocity range (1.03–6.17 m/s) and this could justify why the drag reduction has been increased. Figure 13 depicts this transition clearly.

The formulation of slug flow pattern is always accompanied by a formation of two components (gas pocket and liquid film). As it can be seen from Figure 13a that the gas pocket (at gas liquid interface) penetrates in the stratified liquid film causing an increase in turbulence intensity. Adding the drag reducing polymer is believed to reduce these penetrations, suppresses turbulence patches and also enlarges the stratified liquid film region (Figure 13b). Daas et al. [16] pointed out similar explanation for the drag reduction mechanism in slug flow regimes. Figure 14 is also showing the transition at higher superficial liquid and gas velocities.

### 3.3. Correlations for Gas –Liquid Flow with Addition of DRP

An experimental study has been carried out for a horizontal pipeline to examine the addition of water-soluble polymer on the two-phase water-air flow. The experimental data has been generated based on the two phase (air-water without DRP) map in Figure 15.

In this study, correlations have been developed to allow further understanding of the drag reducing polymers in reducing the frictional pressure gradient and the parameters that could be affected by these additions also explained.

The mixture friction factor $f_{M}$ and the mixture Reynolds number ReM for the two-phase water-air flow are playing a key role in developing good relations that predict and represent the experimental data more properly. The definitions of the mixture friction factor and mixture Reynolds number has been illustrated in the studies of García et al. [18].

#### 3.3.1. Correlation Development

Usually the two-phase water-air is very complex in nature and this complexity is more obviously when the detecting of a flow pattern that could be existed before and after adding the DRP is required. Therefore, the need for developing a correlation that appropriately relate different flow parameters and characteristics receives high attention especially in predicting the pressure drop in pipelines without knowing the flow regime. Possible dimensionless parameter could be the one includes various parameters such as the mixture Reynolds number ReM which comprises pipe diameter, density, viscosity and mixture velocity in one dimensionless number.

Several studies has been performed to correlate the two phase using dimensionless groups, for example García et al. [18,19] developed a correlation of friction factor that covered a wide range of laminar and turbulent flow of gas-liquid regimes. The correlation that been produced was based on liquid holdup ranges to differentiate between the experimental data used in their analysis. However, these correlations have been carried out without the addition of the drag reducing polymers, and hence different trends and correlations could be realized as the DRP added to the system.

Alsarkhi et al. [7] Studied the effectiveness of two correlations for predicting the effect of the drag reducing polymers on the mixture friction factor using published experimental data of air-liquid and oil-water flows in literature. This was the only attempt been found in the open literature at least for predicting the drag reduction in different pipe diameters namely from 0.019 to 0.0953 m.

In the present work, experiments on water-air flow were conducted and several correlation has been developed based on various water superficial velocities using different dimensionless groups and parameters that used in Al-sarkhi et al. [7] and García et al. [18].

#### 3.3.2. Dimensionless Parameters

The mixture friction factor for water–air mixture without the addition of DRP $\left(f_{\mathrm{Mwithout-DRP}}\right)$ is expressed as follows:(1)$$f_{\mathrm{Mwithout-DRP}}=\frac{2\times D\times \frac{dP}{dL}{|}_{withoutDRP}}{\rho_{Μ}\times V_{M}^{2}}$$
where D is the diameter of the pipe, and $V_{M}$ is the mixture velocity which is defined as the summation of liquid and gas superficial velocities ($V_{M}=V_{sl}+V_{sg}$).

The superficial liquid and gas velocities are calculated using the below equations:(2)$$V_{sl}=\frac{4Q_{l}}{\pi D^{2}}$$
(3)$$V_{sg}=\frac{4Q_{g}}{\pi D^{2}}$$
where $Q_{l}$, $Q_{g}$ are the flow rate for the liquid and the gas, respectively.

The mixture density ($\rho_{Μ}$) is defined as:(4)$$\rho_{Μ}=\rho_{l}\lambda_{l}+\rho_{g}(1-\lambda_{l})$$
where: $\lambda_{l}=\frac{Q_{l}}{Q_{l}+Q_{g}}$ the volumetric flow rate fraction, and the $\rho_{l}$, $\rho_{g}$ are the densities of liquid and the gas, respectively.

The mixture friction factor with the drag reducing polymer being added is formulated using the same parameters on Equation (1) above with only changing the pressure drop to $\frac{dP}{dL}{|}_{DRP}$ which is the one with DRP added to the system.
(5)$$f_{\mathrm{M-DRP}}=\frac{2\times D\times \frac{dP}{dL}{|}_{DRP}}{\rho_{Μ}\times V_{M}^{2}}$$

Reynolds number on this analysis is based on the liquid kinematic viscosity $\nu_{L}$
(6)ReM=VM×DνL

The regression analysis is conducted based on the experimental data obtained at different liquid superficial velocities ranged from 1.85 to 4.317 m/s forming around 100 data set points. As it can be seen from Figure 16, that all of the data points are following the same trend of the fitted curve (Equation (7) presents the correlation).

The scatter data conclude a wide range of flow types and regimes of slug (pseudo slug), annular and dispersed bubbly flow regimes. This could enable better prediction of the correlation under the study.

Figure 17 exhibits the comparison between the measured values of friction factor and the predicted one, however as shown in the plot all scatter data has been predicted within band of ± 15%, and the correlation for the friction factor can be represented as:(7)f(M-DRP)=0.0276(ReM(VsgVsl)(0.5))(−0.079)

Using the same experimental data, we were able to generate another correlation that fits the data points exponentially. Since the frictional pressure drop will increase as more liquid flow rate added to the flow; then it could be more interesting to describe such dimensionless pressure drop that includes the frictional pressure gradient when the flow assumed to be liquid only (*P_sl_*) and the pressure drop with the addition of the drag reducing polymer ($P_{DRP}$).

Where $P_{sl}$ defined as:(8)$$P_{sl}=\frac{f\times \rho_{l}\times V_{sl}{}^{2}}{2D}$$

The friction factor (f) for single phase liquid shown in Equation (8) is calculated using the well-known Blasius equation (Equation (9)):(9)f=0.184Resl−0.2

And Resl is Reynolds number that can be expressed as:(10)Resl=ρl×Vsl×Dμl

Figure 18 presents the relation between the dimensionless pressure drop ratio $\frac{P_{DRP}}{P_{sl}}$ and the normalized superficial velocity $\frac{V_{sg}}{V_{sl}}$ As it can be seen that the superficial gas and liquid velocity are in great impact on the pressure drop ratio and could confirm that it is controlling the drag reduction and that makes the correlation behaves better.

The importance of such correlation appears when the Drag Reducing polymers are added to the two-phase system in order to broaden the understanding of the reduction mechanism by using general descriptive model. Using this correlation, we can predict the pressure drop after adding the DRP without knowing the flow pattern and this is quite useful for industrial applications.

The regression analysis is performed for the data points and it can be seen from Figure 19 that all of the scattered data are now in between ±10% spread with correlation goodness of fit $(R^{2}=0.97$). The correlation can be expressed as:(11)$$\frac{P_{DRP}}{P_{sl}}=\left(0.5648\right)\exp\left(0.6456{\left(\frac{V_{sg}}{V_{sl}}\right)}^{0.5}\right)$$

## 4. Conclusions

It can be concluded that the drag reduction in two phase of air-water mixture occurs at all liquid and gas flow rates. The mechanism of drag reduction can be explained as that the DRP leads to suppress the interfacial friction; the polymer solution stretched along the interface to increase laminar sub-layer thickness and result in more unidirectional free of eddies flow.

Utilizing the DRP with low concentrations attributed in pressure drop reduction for all cases been studied, the effectiveness of the DRP varied tremendously from 14% to 80% as reported at the stratified and intermittent flows, respectively.

The DRP can affect flow behavior and found to be more efficient in suppression of highly disturbed waves and found to be able to shift the flow from slug and pseudo slug to stratified wavy and wavy annular regimes, respectively.

In order to produce high drag reduction, the DRP should be able to damp turbulence intensity and fluctuations. The maximum drag reduction always occurs when the highly fluctuated waves were reduced effectively by DRP and accordingly phase transition appeared. For example, DR% of 63% is reported when the slug flow altered to stratified wavy regime with the presence of only 40 ppm DRP.

As discussed previously, the DRP injection mechanism is very sensitive. Therefore, in this study using a diaphragm pump (gives DRP solution in dosages) offered an effective technique by introducing the DRP without causing any polymer shear degradation. Moreover, with an easy adjustable flow rate controller attached to the pump this method of injection can give a reliable way for industrial applications, since it can provide a wide range of flow rates with changeable speeds and torques by which the exact needed amount of polymer solution would be controlled more accurately.

Friction factor correlation as a function of mixture Reynolds number obtained in this study is evaluated and it has been successfully covered a wide range of liquid and gas flow rates (including different flow regimes) and predicted the data effectively within ±15% when it is compared with the measured values.

The pressure drop after the addition of DRP $(P_{\mathrm{DRP}}$) has been predicted using the experimental data, the correlation has been presented as a function of the superficial frictional pressure drop and normalized superficial velocity. Regression analysis is conducted over a wide range of the experimental data, the correlation effectively predicted the data within ±10% spread with correlation goodness of fit ($R^{2}=0.97)$.

## Figures and Tables

**Figure 1 polymers-15-01108-f001:**
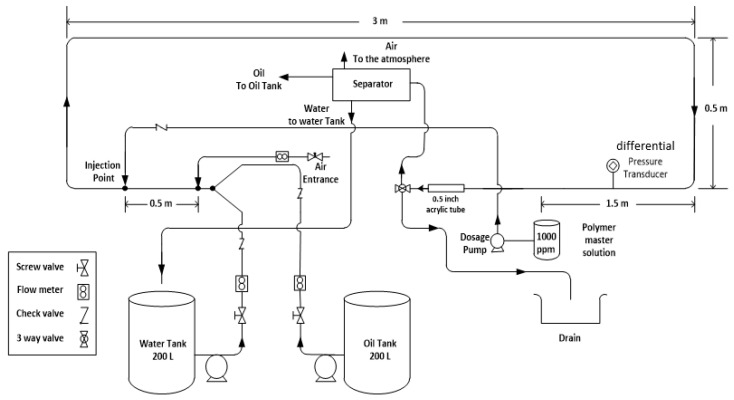
Sketch of the flow facility.

**Figure 2 polymers-15-01108-f002:**
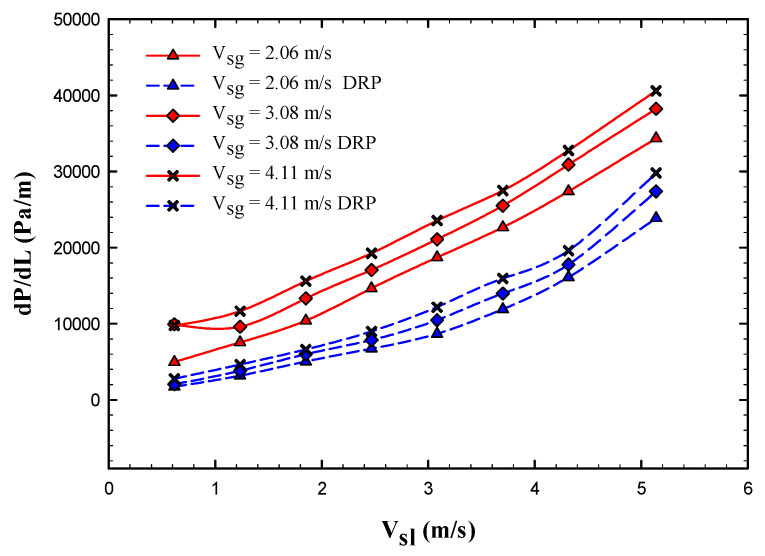
Comparison of the frictional pressure drop variation with respect to liquid superficial velocity at different gas superficial velocities of 2.06, 3.08 and 4.11 m/s.

**Figure 3 polymers-15-01108-f003:**
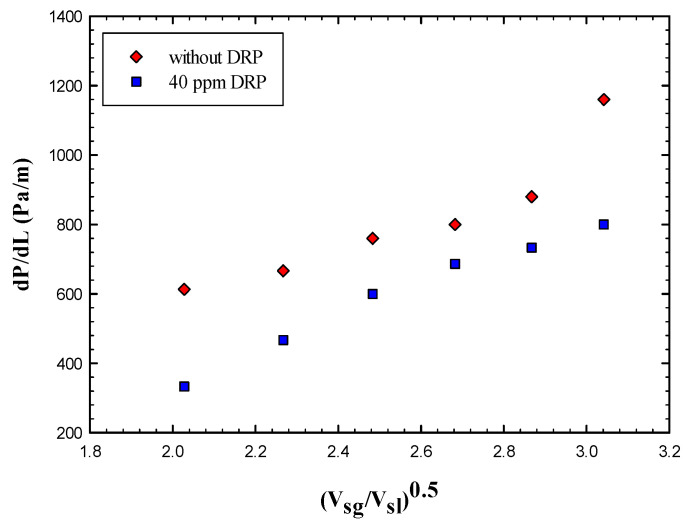
Effect of drag reducing polymer on the stratified flow regime.

**Figure 4 polymers-15-01108-f004:**
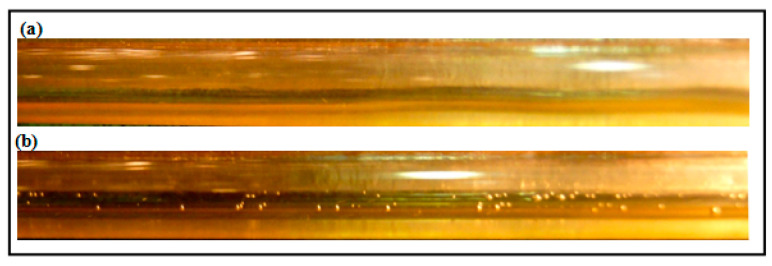
(**a**) Stratified Wavy flow without DRP (V_sl_ = 0.1 m/s, V_sg_ = 0.41 m/s); (**b**) Stratified Wavy flow with 40 ppm DRP.

**Figure 5 polymers-15-01108-f005:**
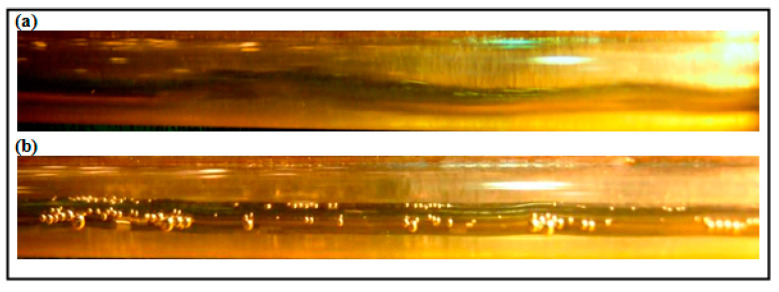
(**a**) Stratified Wavy flow without DRP (V_sl_ = 0.1 m/s, V_sg_ = 2.88 m/s); (**b**) Stratified Wavy flow with 40 ppm DRP.

**Figure 6 polymers-15-01108-f006:**
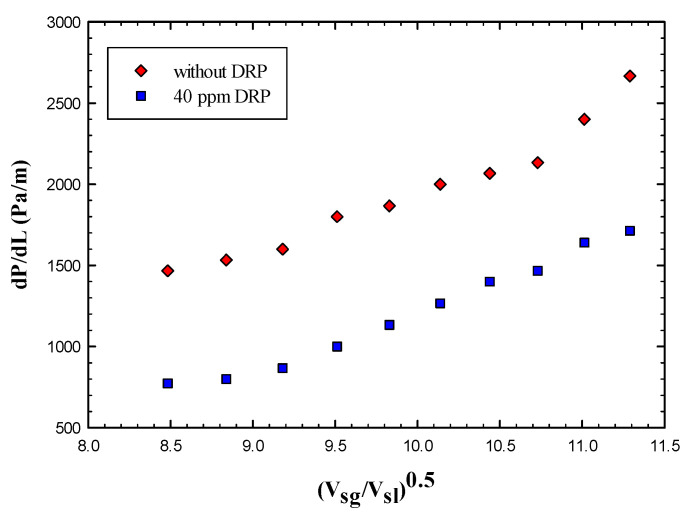
Effect of drag reducing polymer on annular flow regime.

**Figure 7 polymers-15-01108-f007:**
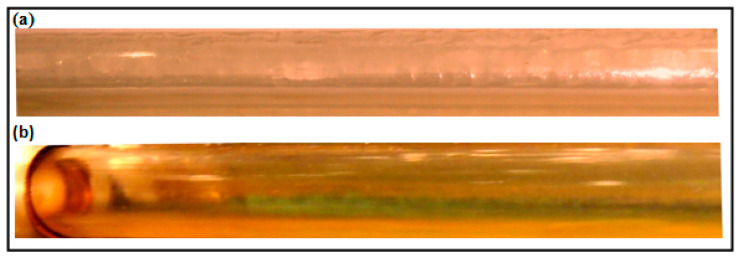
(**a**) Annular flow regime without DRP (V_sl_ = 0.1 m/s, V_sg_ = 9.05 m/s); (**b**) Annular flow regime with 40 ppm of DRP.

**Figure 8 polymers-15-01108-f008:**
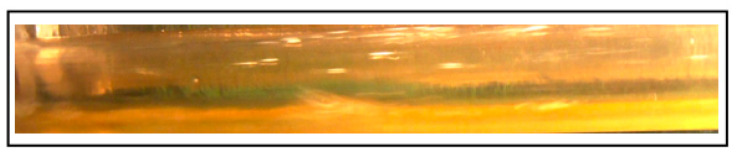
Annular wavy flow regime (V_sl_ = 0.1 m/s, V_sg_ = 12.75 m/s).

**Figure 9 polymers-15-01108-f009:**
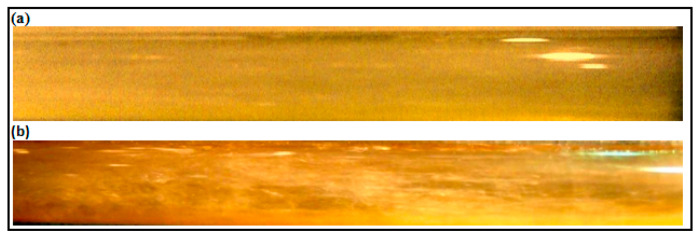
(**a**) Typical feature of a Dispersed Bubbly flow regime (V_sl_ = 3.08 m/s, V_sg_ = 1.03 m/s) (**b**) Transition from Dispersed Bubbly to Pseudo slug flow regime with 40 ppm DRP.

**Figure 10 polymers-15-01108-f010:**
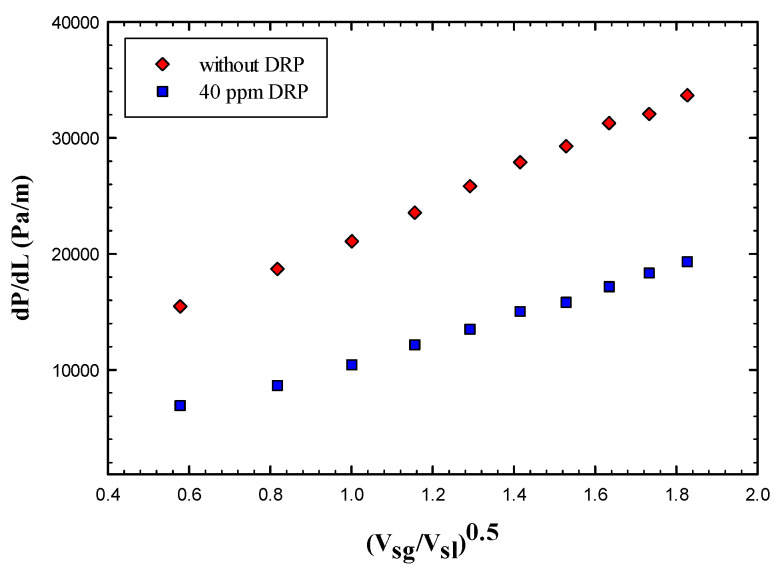
Effect of drag reducing polymer on dispersed bubbly flow regime.

**Figure 11 polymers-15-01108-f011:**
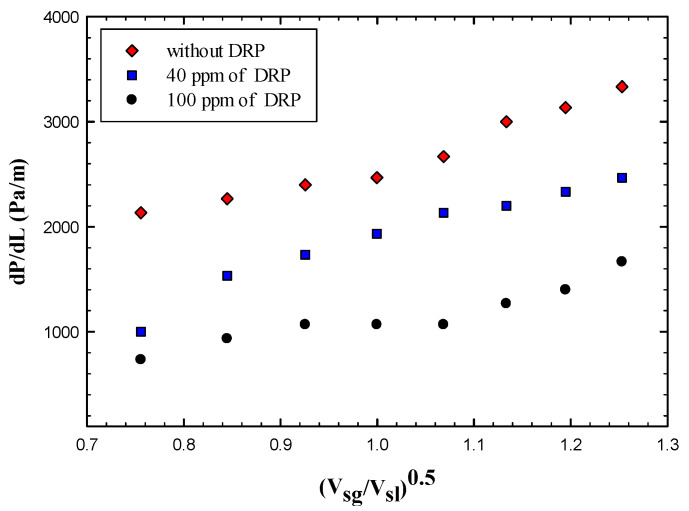
Effect of drag reducing polymer on slug flow regime using a concentration of 40 and 100 ppm.

**Figure 12 polymers-15-01108-f012:**
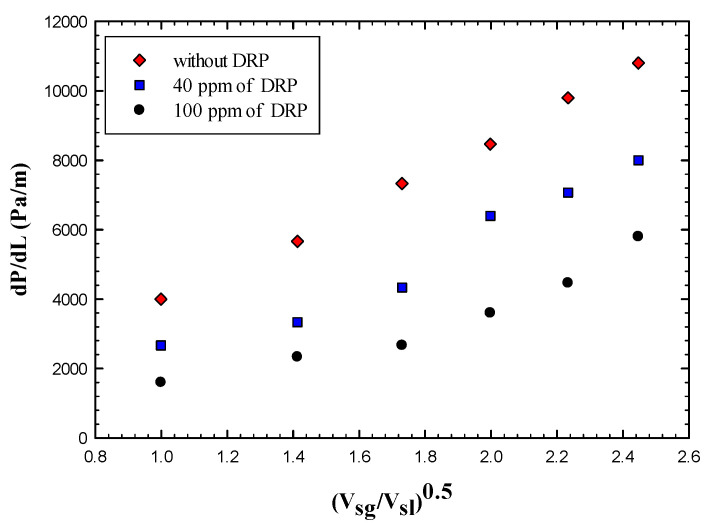
Effect of drag reducing polymer on pseudo slug flow regime using a concentration of 40 and 100 ppm.

**Figure 13 polymers-15-01108-f013:**
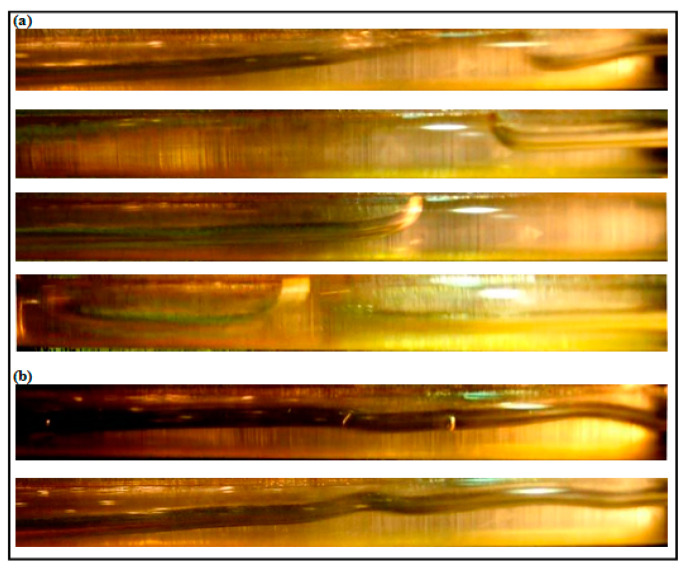
(**a**) Slug flow regime without DRP (V_sl_ = 0.72 m/s, V_sg_ = 0.41 m/s); (**b**) Transition from Slug to Stratified Wavy flow regime using 100 ppm DRP.

**Figure 14 polymers-15-01108-f014:**
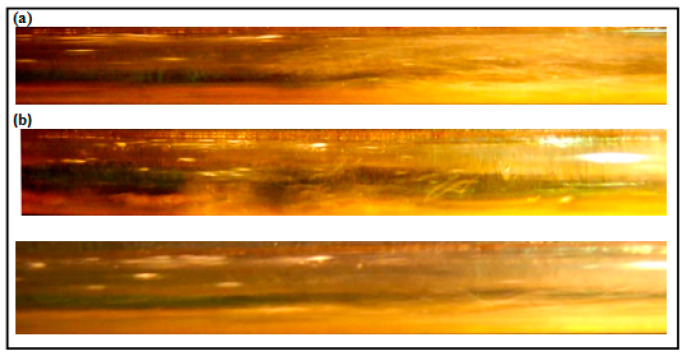
(**a**) Pseudo Slug flow without DRP (V_sl_ = 1.03 m/s, V_sg_ = 3.08 m/s); (**b**) Transition from Pseudo Slug to Wavy Annular flow regime using 100 ppm.

**Figure 15 polymers-15-01108-f015:**
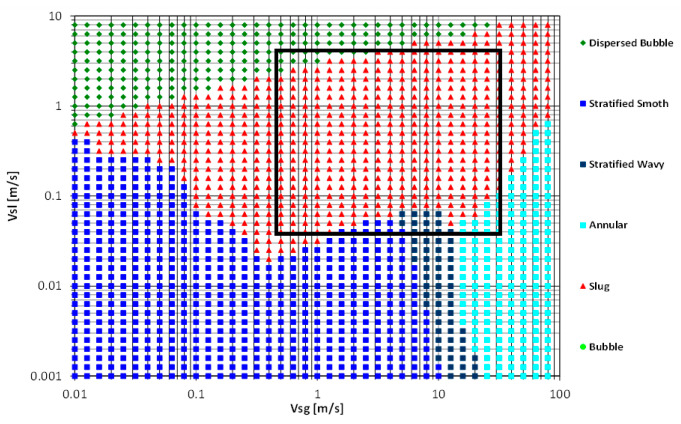
Air-water (without DRP) flow pattern map using Unified [17] Model in a 1.01-cm pipe. (Dashed box is the present work flow conditions) Where: **DB:** dispersed bubble, **SL:** Slug, **IN:** Intermittent, **SS:** Smooth Stratified, **SW:** Stratified Wavy, **AN:** Annular.

**Figure 16 polymers-15-01108-f016:**
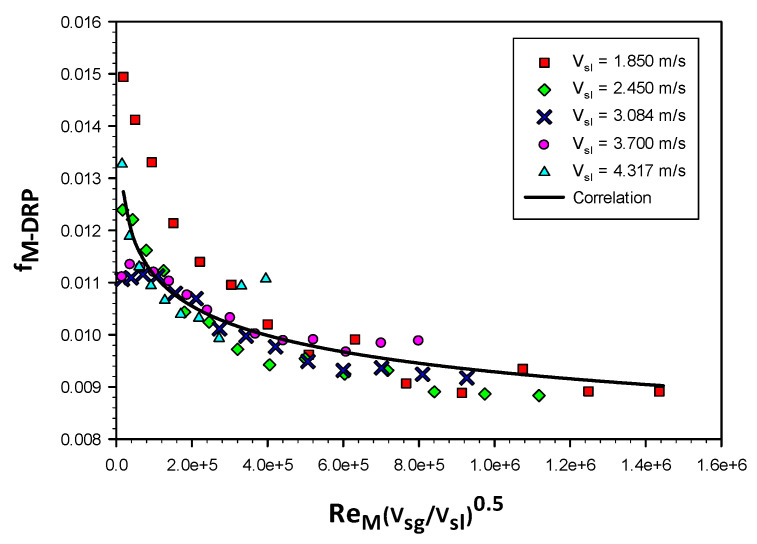
Friction factor variation with the mixture Reynolds number times the square root of the superficial velocities ratio for different liquid superficial velocities (1.85, 2.45, 3.08, 3.7 and 4.32 m/s).

**Figure 17 polymers-15-01108-f017:**
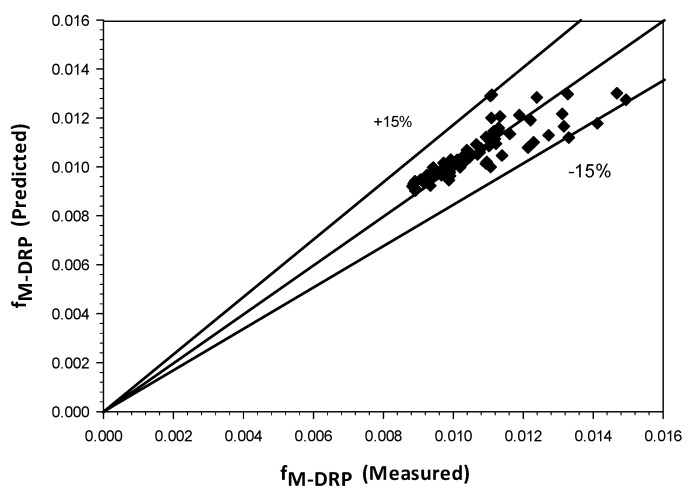
Comparison between measured friction factor and predicted by Equation (7).

**Figure 18 polymers-15-01108-f018:**
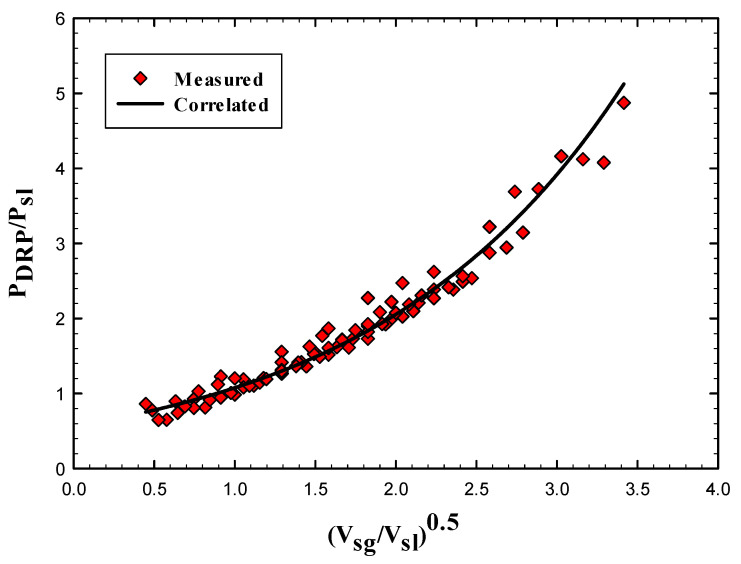
Dimensionless pressure drop ratio versus square root of the normalized superficial velocities (correlation is Equation (11)).

**Figure 19 polymers-15-01108-f019:**
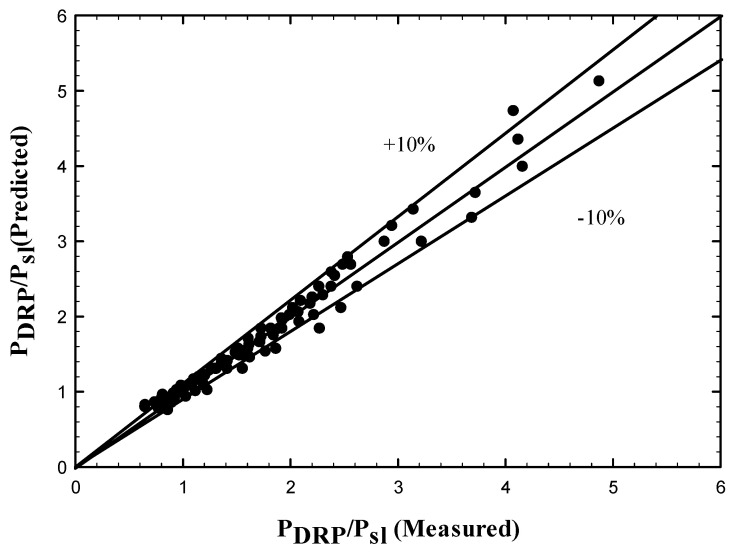
Comparison between measured dimensionless pressure drop ratio and the predicted by Equation (11).

**Table 1 polymers-15-01108-t001:** Fluids standard properties.

Water Density	$1000\;\frac{\mathrm{kg}}{m^{3}}$
Water viscosity	0.000891 Pa s
Ph	7–8
Gas Density	$1.28\;\frac{\mathrm{kg}}{m^{3}}$
Gas viscosity	0.0000185 Pa·s

**Table 2 polymers-15-01108-t002:** Polymer technical properties from manufacturer.

Product Name	Coopolymer of Acrylamide and Quaternized Cationic Monomer		
Product Type	Powder		
physical form	off-white granular solid		
cationic charge	Medium-high		
Molecular weight	very high		
specific gravity	0.75		
Bulk density	749.66 kg/m^3^		
Ph 1% solution	4–6		
Apparent Viscosity/(cP) 25 °C			
Concentration	0.0025	0.005	0.01
Viscosity	650	1200	3000

## Data Availability

All the data is presented in the article.

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
