# Peer review of "Multiphase Flow Production Enhancement Using Drag Reducing Polymers"

_polymers, 2023, doi:10.3390/polym15051108_

Round 1

Reviewer 1 Report

The article describes an experimental study on the effect of drag reducing polymers (DRP) on a horizontal pipe carrying a two-phase flow of air and water mixture. The study aims to examine the ability of these polymers to reduce drag, dampen turbulence waves, and change the flow regime in various conditions. The paper needs to be significantly improved before publication. The authors need to address the following issues:

1. In the last paragraph of the introduction the authors need to explain what are they doing in the paper and what is the novelty of the work

2. Viscosity in table 2 is of an aqueous solution of the polymer and not of the polymer. It should be in a different table. The units of the concentration are missing.  40 ppm and 100 ppm of DRP are used in the experiments. What is the viscosity of these solutions?

3. All units should be in the metric system.

4. Information in table 3 should be in a graph only. The paper repeats the same data in different places. It gets confusing for the reader.

5. The authors explain drag reduction by the following:

>*The mechanism is that the DRP lead  to suppress the interfacial friction; the polymer solution stretched along the interface to increase laminar sub-layer thickness and result in more unidirectional free of eddies flow

Since the interface behaviour is affected by interfacial tension the authors need to check how the DRP influences interfacial tension and eventually if some of the results can be explained by Marangoni effect.

6. The English needs to be significantly improved.

7. What is the theoretical basis for the correlations?

8. The paper has a lot of intermediate data that would be better to not appear in the final paper. For example, correlation used in fig 16 is not the final one. It would make more sense to apply eq. 7 to the data and present only the data fitting to eq 7. Fig 18 is also not the one representing the final correlation.

9. the correlation used in fig 18 should appear in the plot. The same in figure 16.

10. What is the source of equation 9? Why the need to introduce this equation?

11. It would be better for the paper to have a theory section where  the main correlations studied are presented and the different regimes are explained. I would include figure 15 in this section.

12. The authors should consider move some of the tables of the results to supplementary material.

Author Response

  1. In the last paragraph of the introduction the authors need to explain what are they doing in the paper and what is the novelty of the work

Answer: the following paragraph was added in the last paragraph

The novelty of the present work is that the experiments were conducted in a 1.016-cm ID tube in which the experimental data is rarely exited in open literatures. Moreover, different flow regimes were tested and a new, empirical correlation is developed which enables predicting the pressure drop after the addition of DRP to give an estimate of the amount of drag reduction.

  1. Viscosity in table 2 is of an aqueous solution of the polymer and not of the polymer. It should be in a different table. The units of the concentration are missing.  40 ppm and 100 ppm of DRP are used in the experiments. What is the viscosity of these solutions?

Answer: The viscosity is the same as water. The 100 ppm did not affect by any means the viscosity or the density of the aqueous solvent (water)

3. All units should be in the metric system.

Answer :Done

4. Information in table 3 should be in a graph only. The paper repeats the same data in different places. It gets confusing for the reader.

Answer: I will move it to Appendix. I prefer keeping the row data for the benefit of the readers to use it directly instead of getting the values from a graph.

5. The authors explain drag reduction by the following:

>*The mechanism is that the DRP lead  to suppress the interfacial friction; the polymer solution stretched along the interface to increase laminar sub-layer thickness and result in more unidirectional free of eddies flow

Since the interface behaviour is affected by interfacial tension the authors need to check how the DRP influences interfacial tension and eventually if some of the results can be explained by Marangoni effect.

Answer: again same as density and viscosity the 100 ppm does not change any properties it acts mechanically the entanglements of the DRP stretch absorbing the turbulent energy then killing the vortices not by changing the properties of the solvent

6. The English needs to be significantly improved.

Answer: done thank you

7. What is the theoretical basis for the correlations?

The basis is just trend line of the data and the proper dimensionless parameters used in the correlation

8. The paper has a lot of intermediate data that would be better to not appear in the final paper. For example, correlation used in fig 16 is not the final one. It would make more sense to apply eq. 7 to the data and present only the data fitting to eq 7. Fig 18 is also not the one representing the final correlation.

Answer: no all of them are usable correlation but using different dimensionless parameters with different percentage error. In fact in multiphase flow area sometimes they relate the two phase pressure drop to the pressure drop of single phase liquid this correlation is in line with this procedure.

9. the correlation used in fig 18 should appear in the plot. The same in figure 16.

But they are different, one of them in Figure 16 correlating the friction factor and the one in figure 18 is correlating the pressure drop with polymer in the two-phase flow to the single phase liquid pressure drop this is familiar in two phase flow

10. What is the source of equation 9? Why the need to introduce this equation?

Answer: that is the well-known Blasius equation for friction factor of pure single-phase fluid flow in pipe

11. It would be better for the paper to have a theory section where  the main correlations studied are presented and the different regimes are explained. I would include figure 15 in this section.

Answer: I really prefer to have it in the results section because it is really a way of presenting the results and get benefit or generalize it by making it dimensionless

  1. The authors should consider move some of the tables of the results to supplementary material.

Answer: I moved all tables of results  into appendix section at the end

Reviewer 2 Report

The authors investigated the pressure drop in the multiphase flow using drag-reducing polymers, which is vital for the multiphase flow. The following are my comments:

1.       There are too many typos and grammar errors in the draft.

2.       Fig. 1 shows there is only one pressure transducer, then how do get the pressure drop?

3.       Tables 1 and 2 need to be reformed into three-line tables.

4.       In table 2, the molecular weight is very high. How high is “very high”? Please give the value.

5.       As water and oil form the liquid phase, what are the flow rates of water and oil, respectively, in each case? Whether the proportion of water and oil affects the results?

6.       In Fig. 15, what is the polymer concentration?

7.       As the polymer concentration influence the results shown in Figs. 11 and 12, why there’s no parameter relevant to concentration in the correlation equations? How do the volume fractions of water, oil, and air affect the friction factor?

Author Response

The following are my comments:

  1. There are too many typos and grammar errors in the draft.

Answer: the paper has been revised now

  1. 1 shows there is only one pressure transducer, then how do get the pressure drop?

Answer: No it is a differential pressure transducer the word differential has been added to the figure and the length over which the pressure drop is measured is shown clearly in the figure 1.5 m. if fact in the setup we have two differential pressure transducers one measures over a 1 m length ( close to entrance) and one closes to the end and measure along 1.5 m length.

  1. Tables 1 and 2 need to be reformed into three-line tables.

Answer: Done

  1. In table 2, the molecular weight is very high. How high is “very high”? Please give the value.

Answer: this is what is written by the manufacturer but it is known in this type of research that high is usually higher than a Million to be able to see drag reduction

  1. As water and oil form the liquid phase, what are the flow rates of water and oil, respectively, in each case? Whether the proportion of water and oil affects the results?

Answer: In this paper we used only air as a gas phase and water as a liquid phase

  1. In Fig. 15, what is the polymer concentration?

Answer: This is at zero ppm just for air and water and it is now added in the revised version as

Figure 15. Air-water (without DRP) flow pattern map using Barnea [13] Model in a 0.4 inch pipe.”

  1. As the polymer concentration influence the results shown in Figs. 11 and 12, why there’s no parameter relevant to concentration in the correlation equations? How do the volume fractions of water, oil, and air affect the friction factor?

Answer: all reported results are at maximum drag reduction point i.e we increase the concentration from zero until no more drag reduction is observed then we report the pressure drop. Beyond the maximum drag reduction adding more polymers will not change any thing. Usually you can see drag reduction at very small ppm in few ppm but people concern about the maximum drag reduction 

Round 2

Reviewer 1 Report

The authors did not address some of the issues 2, 5, 8, 9, 10 and 11. In particular, for 2, 5 and 10:

2. What is the viscosity in table 2 and why is there?

5. Do you have any proof that DRP is not acting like a sufactant?

10. What is the need for the equation?

Author Response

The authors did not address some of the issues 2, 5, 8, 9, 10 and 11. In particular, for 2, 5 and 10:

2. What is the viscosity in table 2 and why is there?

This is taken directly from data sheet of the DRP producer. It gave the viscosity at three different concentrations. For example at the lowest given concentration of 0.0025 the viscosity is 650 cp. If you want to convert the 0.0025 into PPM (parts per Million) it will be 2500 ppm which is way above the ppm used in this study that is way we can say with confidence that the 100 ppm maximum used DRP in water will not alter the density neither will change the viscosity or the surface tension. This paragraph has be added now in the revised version “Table 2 shows data from the manufacturer product sheet indicating the viscosity of the DRP at the specific concentration. It is worth to be mentioned here that at 100 PPM DRP in water which is the maximum concentration used in the present experiments neither the viscosity of the water with 100 ppm DRP nor the density or surfactant will be affected by the DRP presence in such a small amount. “

5. Do you have any proof that DRP is not acting like a sufactant?

It is well known that the DRP at this little concentration ( 100 ppm) does not change the surface tension of the water at all at the concentration used. I will revised the paper and write it clearly in the revised paper.

10. What is the need for the equation?

This equation (equation 9)  is needed to calculate the friction factor implemented in  equation 8. I wrote it clearly now in the revised version. As “ The friction factor (f) for single phase liquid shown in equation 8 is calculated using the well-known Blasius equation (equation 9):